# Public Awareness of Chronic Kidney Disease in Jazan Province, Saudi Arabia—A Cross-Sectional Survey

**DOI:** 10.3390/healthcare10081377

**Published:** 2022-07-25

**Authors:** Ali Assiry, Saeed Alshahrani, David Banji, Otilia J. F. Banji, Nabeel Kashan Syed, Saad S. Alqahtani

**Affiliations:** 1Department of Pharmacology and Toxicology, College of Pharmacy, Jazan University, Jazan 45142, Saudi Arabia; ali.shar999@gmail.com (A.A.); davidbanji@gmail.com (D.B.); 2Mohayil Hospital, Health Affairs of Aseer, Abha 62523, Saudi Arabia; 3Department of Pharmacy Practice, College of Pharmacy, Jazan University, Jazan 45142, Saudi Arabia; otiliabanji@gmail.com (O.J.F.B.); nsyed@jazanu.edu.sa (N.K.S.); ssalqahtani@jazanu.edu.sa (S.S.A.); 4Pharmacy Practice Research Unit, College of Pharmacy, Jazan University, Jazan 45142, Saudi Arabia

**Keywords:** chronic kidney disease, knowledge, awareness, risk factors, complications

## Abstract

Chronic Kidney Disease (CKD) is a significant public health concern worldwide and many people continue to ignore their early warning symptoms. The present study assessed the level of knowledge about CKD the awareness of the risk factors and the awareness of the complications associated with CKD, among the general population of Jazan Province, Saudi Arabia. 440 residents of Jazan Province participated in an online cross-sectional survey during a seven-month period from November (2020) to July (2021). Data was collected using a validated 73-item self-report survey. More than half of the respondents were males (n = 286; 65%) with an age ranging from 18 to 59 years, and a mean age of 32.66 years (SD ± 10.83). A very low percentage of the sample (27.3%; 7.5%, 9.3%) demonstrated good knowledge, a high level of awareness of the risk factors, and a high level of awareness of the complications associated with CKD, respectively. Participants’ knowledge was significantly associated with being a student or being employed (Government/private employee) (χ^2^ = 29.90; *p* < 0.001), having completed graduate studies (χ^2^ = 63.86; *p* < 0.001), residing in urban areas (χ^2^ = 138.62; *p* < 0.001), belonging to the age group (18–39 years), and having no co-morbidities (χ^2^ = 24.55; *p* < 0.001). Positive and significant correlations were also noted between the knowledge score and the awareness of risk factor score (r = 0.42; *p* < 0.01), as well as the awareness of complications score (r = 0.25; *p* < 0.01). These findings warrant an urgent need for extensive CKD educational initiatives concentrating on improving the general knowledge and awareness of the public towards CKD.

## 1. Introduction

Chronic kidney disease (CKD) is a growing public health problem and a significant burden on the health care system. In 2017, the prevalence of CKD globally was estimated to be 9.1%, with 6975 million affected [1]. The Global Burden of Disease study emphasizes that CKD ranks 12th among the causes of death worldwide and in the top five causes of death in many countries [1], with an estimated prevalence of 15.8% [2,3,4,5]. The age-standardized prevalence of CKD in Saudi Arabia is relatively high, with an estimated 9892 per 100,000 affected [6,7]. A 2–4-fold higher prevalence of CKD was found in relatives of patients with CKD in Southwestern Saudi Arabia [6,7]. CKD can serve as a significant risk factor for stroke and ischemic heart disease [1].

According to Kidney Disease: Improving Global Outcomes (KDIGO), CKD is defined as abnormalities that are related to the kidney structure or function, have been present for more than three months, and have implications for an individual’s health [8]. CKD is also defined as a state in which the estimated glomerular filtration rate < 60 mL/min/1.73 m^2^, along with persistent albuminuria [8], as well as the loss of 50% of kidney function from stage 3 to stage 5 with a significant risk of morbidity and premature death [9,10,11].

CKD is especially problematic for many who are more likely to get the disease at earlier ages [9,10,11,12]. They also progress more quickly to End-Stage Renal Disease (ESRD), the final stage of CKD, treatable only by lifelong dialysis or kidney transplant [9,10,11,12]. Treatment comes with high costs in terms of public and private money and patients’ suffering [13]. Early detection and treatment of CKD are essential to prevent disease progression [11,12,13]. Unfortunately, most people with CKD are undiagnosed and hence will be undertreated, leading to disease progression as well as complications. Diabetes, hypertension, and a family history of kidney disease are the most significant risk factors associated with CKD and, owing to these aforesaid reasons, it is highly recommended that adults with any of these should be thoroughly screened [12,13,14].

Considering the high age-standardized prevalence of CKD (i.e., 9892 per 100,000) in Saudi Arabia, in comparison with Europe and North America, only a few studies have been conducted in this region amongst the general population to assess their knowledge and awareness regarding CKD [6,7]. Hence, the present study was designed and conducted to assess the knowledge about CKD and also to assess the awareness of the risk factors as well as the complications associated with CKD amongst the general population of Jazan Province, given the knowledge that no such study has been previously conducted in these demographics.

## 2. Materials and Methods

### 2.1. Study Setting and Participants

Data was collected using an online cross-sectional survey. The self-report questionnaire was prepared and hosted on *Google Forms*. An invitation link to participate in the survey was circulated on a range of diverse social media platforms (*WhatsApp*, *Facebook*, *Instagram*, and *Twitter*) to various groups across the different provinces of Saudi Arabia. It was highly encouraged to share the invitation link among participants’ professional, as well as personal, contacts. Additionally, frequent reminders were also sent to convince the participants to return completed surveys. Snowball convenience sampling was used to recruit the participants on social media [10]. Data collection was carried out for a duration of seven months from November (2020) to July (2021). The web-based survey was completed by Saudi resident males and females. The final sample consisted of 440 participants, of which more than half were males (n = 286; 65%) and the rest were females (n = 154; 35%). Participants had ages ranging from 18 to 59 years, and a mean age of 32.66 years (SD ± 10.83).

### 2.2. Inclusion and Exclusion Criteria

Participants to be included in the present study were required to (i) be Saudi resident males or females, (ii) be at least 18 years of age, (iii) have an appropriate understanding of English (and/or) Arabic languages [13], (iv) be willing to participate in the present study, as well as (v) be willing to provide informed consent. Those individuals not fulfilling the aforementioned inclusion criteria were excluded from the study [14].

### 2.3. Data Collection Tools

A self-report 73-item questionnaire was the tool for data collection. The self-report survey comprised 4 sections. Section 1 comprised 9 questions related to participants’ demographics as well as medical history. Section 2 comprised 18 questions assessing the participants’ knowledge about CKD. Section 3 comprised 25 questions assessing the participants’ awareness of risk factors for CKD. Section 4 had 20 questions evaluating the participants’ awareness of the complications of CKD.

### 2.4. Survey Instrument Development, Validation, Translation, and Pilot Study

The first draft of the questionnaire was prepared by the authors after careful, detailed, and protracted deliberation and review of the current literature [1,2,3,4,5,6,7,8]. A four-member independent expert questionnaire review committee was constituted, comprising two experienced physicians (nephrologists) and two clinical pharmacists (with nephrology experience) to review the survey instrument for face and content validation. The questions deemed unsuitable by the expert review committee were consequently deleted. The final, approved draft of the study questionnaire consisted of 73 items, divided into 4 sections (Demographics, Knowledge about CKD, Awareness of risk factors associated with CKD, and Awareness of the complications associated with CKD).

The final draft of the English version of the questionnaire was translated into Arabic using the forward-backward translation technique [14,15,16]. An independent professional translator with bilingual expert-level proficiency in both Arabic and English was used to translate the final approved English version of the questionnaire into Arabic. This was then reviewed by an author having expert-level bilingual proficiencies. The presence of any discrepancies was then discussed with the independent translator and resolved. A final version was then prepared and approved. The approved final version of the Arabic questionnaire was then back-translated into English by another study author who until then had no prior knowledge of the English version. Lastly, the forward, as well as the backward translated versions, were reviewed by all the study authors having bilingual proficiencies [15,16].

The final version of the study instrument was then presented to an independent focus group of 40 participants to evaluate the completion time as well as the ease of use [14,15,17,18,19]. The survey questions were easily understood by the participants of the pilot study and an average completion time of 14 min was recorded. The pilot study sample was used to test the validity and reliability of the study instrument. Face and content validation was assessed, while the reliability was assessed using Cronbach’s alpha. The independent pilot study sample was subsequently excluded from the final analysis.

#### Reliability Analysis

For assessing the internal consistency of the different items of different sections of the scale, Cronbach’s alpha was used. Alpha coefficients for all the items of the different sections of the scale were found to be above 0.87 (i.e., greater than the 0.70 threshold), demonstrating excellent reliability of the developed scale [19,20,21,22]. The alpha coefficients of the different sections of the study questionnaire are as follows: Participants’ knowledge about kidney problems and CKD (α = 0.90); Participants’ awareness of risk factors associated with CKD (α = 0.87); Participants’ awareness of complications associated with CKD (α = 0.88).

### 2.5. Measures

#### 2.5.1. Socio-Demographic Information

Participants in the present study reported their gender, age group, exact age, nationality, education level, occupation, location of current residence, and medical history of non-communicable and communicable diseases.

#### 2.5.2. Participants’ Knowledge about Kidney Problems and CKD

The second section comprised 18 closed-ended questions designed to assess participants’ knowledge about CKD. Participants were required to provide their responses by selecting one of the three options: ‘yes’, ‘no’, or ‘I don’t know’. An item exemple includes “*CKD can cause retention of fluid.*” the items of which are scored from 1 (yes) to 0 (no) or I don’t know) (Appendix A, see Appendix A). Each accurate response ‘yes’, received one point, whereas ‘no’ or ‘I don’t know’ responses received zero points [23]. The knowledge score was calculated by adding the scores of all the items in the section. Higher scores indicated excellent knowledge about CKD.

#### 2.5.3. Participants’ Awareness of Risk Factors Associated with CKD

The third section comprised 25 closed-ended questions assessing the participants’ awareness of the risk factors associated with CKD. Participants were also required to provide their responses by selecting one of the three options: ‘yes’, ‘no’, or ‘I don’t know’ (Appendix A, see Appendix A). An item example includes “*People over the age of 60 years are at risk of CKD.*”, the items of which are scored from 1 (yes) to 0 (no or I don’t know). The awareness of risk factors score was then calculated by adding the individual scores of each item in the section. Higher scores indicated excellent awareness of risk factors associated with CKD.

#### 2.5.4. Participants’ Awareness of Complications Associated with CKD

Section 4 consisted of 20 closed-ended questions assessing the participants’ awareness of complications associated with CKD. Similar to the other sections, the participants were also required to provide their respective responses by choosing one of the three provided options: ‘yes’, ‘no’, or ‘I don’t know’ (Appendix A). An example item includes “*Fluid build-up is a complication associated with CKD.*”, the items of which are scored from 1 (yes) to 0 (no or I don’t know). The awareness of complications score was assessed by summing up the scores of all items in the section. Higher scores indicated excellent awareness of complications associated with CKD.

### 2.6. Sample Size Calculation

For sample size calculation, the *Raosoft sample size calculator* was used. Based on a 5% margin of error, 95% confidence interval, an approximate population size of 1,000,000, power (1-β) of 0.80, and a 50% response distribution, a sample size of 384 was determined [14,15,24,25,26]. Cochran’s equation [27], was further used for reconfirmation of the calculated sample size.

The determined sample size was then additionally cross-checked and further reconfirmed by making use of Open Epi [27] using a population size of 1,000,000; a finite population correction factor; a confidence limit of 5%; a design effect of 1 and yielding a 384-sample size [28].

### 2.7. Data Collection

The data in the present study was collected using a web-based survey. Volunteer data collectors were recruited using the convenience sampling method by making an announcement in different (College of Pharmacy, Jazan University) student groups asking interested students to contact the corresponding author. The details of the study and awarding of certificates of appreciation as an incentive were also added to the announcement. Participation in the study was entirely voluntary. Ten internship and final year (five male and five female students) volunteered to be data collectors. Each volunteer data collector was required to sign a consent form stating that they were participating in the study of their own free will without receiving any financial incentives. The volunteer data collectors were responsible for sending the invitation link to different social media groups which would ensure a minimum sample size of 40 participants per data collector. The volunteer data collectors were responsible for sending out frequent reminders encouraging participants to complete the survey until the minimum calculated sample of 385 was achieved, after which the invitation link was kept open for another week. The volunteer data collectors were given certificates as official data collectors for the study and also certificates of appreciation. The whole process of data collection was closely monitored by the study authors. Informed consent was obtained by requesting the participants to answer ‘Yes’ to an obligatory question seeking their consent for their willing participation in the study. Providing prior informed consent was essential for the participants to continue to other sections of the questionnaire. A ‘No’ response subsequently ended the survey with the corresponding response being considered a dropout. A respondent’s failure to complete any question (and/or) any section rendered an incomplete response which was consequently excluded from the statistical analysis [11,12]. The completion time of the pilot and the main study were comparable (i.e., 14 min). A total of 500 responses were received out of which only 440 were complete. The response rate of the present study was (440/500) 88%.

### 2.8. Data Analysis

Statistical Package for the Social Sciences (SPSS Inc.; Chicago, IL, USA) (version 23) was used for data analysis. The data from Google forms were entered into Excel sheets and coded and subsequently entered into SPSS. Participants’ socio-demographics were analyzed by making use of descriptive statistics and were later expressed as frequencies, total percentages, means, and standard deviations. Pearson’s correlation coefficient was calculated to test the correlation of continuous variables (i.e., scores of different sections). A multivariate binary logistic regression test was also used for assessing any potential associations between socio-demographics and participants’ knowledge as the outcome variable. A *p <* 0.05 was considered statistically significant.

### 2.9. Ethical Considerations

The present study was conducted only after it was approved by the Institutional Research Review and Ethics Committee of Jazan University (Letter number REC 42/1/127 dated 5 May 2021). Informed consent was taken by requesting the study participants to answer ‘Yes’ to a mandatory question seeking their consent. Providing informed consent was imperative for the study participants to proceed to other sections of the questionnaire. A ‘No’ answer automatically ended the survey, with the corresponding response being considered a dropout.

## 3. Results

Shapiro–Wilk along with Kolmogorov–Smirnov tests were carried out to check the normality of data and the data was found to be normally distributed.

Table 1 depicts the results of the socio-demographics of the participants. It was noted in the present study that more than half of the participants were male (n = 286; 65%) and the remaining female (n = 154; 35%). More than three-quarters of the sample (75.2%) were aged 18–39 years. Almost the entire study sample (91.8%) was Saudi. Most of the participants (72%) were graduates. More than half of the sample (55.9%) were either employed by private firms or had government jobs. Similarly, half of the participants (68.2%) were residents of urban areas. The majority of the study participants did not report the presence of any comorbidities (70.9%). A vast majority of the respondents also did not report communicable diseases (70.9%) or non-communicable diseases (93.4%).

### Scoring of the Questionnaire

The survey comprised three sub-sections covering two domains (i.e., knowledge and awareness). The total score of the sub-section was calculated by adding the individual scores of each item in that sub-section. Following a thorough literature review, it was decided to use Bloom’s cut-off point [28,29] for the categorization of the knowledge and awareness scores. Any score less than 60% of the total score was categorized as poor, while any score in the range of 60–80% was categorized as moderate and, scores greater than 80% of the total score were categorized as good/high (Table 2) [21,27].

Appendix A (see Appendix A) elucidates the participants’ responses to questions assessing their knowledge of kidney problems and CKD. More than half of the participants (58.9%, 76.4%, 70.9%, 77.3%, 78.6%, and 79.3%) had knowledge that many people are living with kidney problems, that the function of the kidney was to filter out waste and water from the blood, that CKD can cause fluid retention, that CKD can lead to an increase in poisonous substances in the body, individuals can survive with one kidney, and that kidney diseases are preventable, respectively, as they responded ‘yes’ to these questions thus reflecting the presence of good knowledge among the participants regarding kidney problems and CKD. On the contrary, a vast majority of the participants (55.5%, 56.8%, 66.8%, and 71.8%) did not know that the kidney filters 200 quarts of blood per day, that an increase in creatinine is always indicative of kidney problems, that most cases of CKD in adults are inherited rather than acquired, and that every 14 days a new name is added to the kidney transplant waiting list, respectively, as they responded ‘no/I don’t know’ to the above questions and in the process demonstrated poor knowledge.

Appendix A (see Appendix A) depicts the participants’ responses to questions assessing their awareness of the risk factors associated with CKD. Nearly three-quarters of participants (75.7%, 75%, 73.2%, and 73.9%) were aware that abuse of pain reliever drugs (NSAIDs) can cause CKD, that dehydration can cause an acute kidney injury (AKI), and that drinking alcohol can cause CKD, respectively, as they responded ‘yes’ to the aforesaid questions thus reflecting good awareness towards risk factors associated with CKD. Contrastingly, a high percentage of participants (80.7%, 78%, 79.1, 79.8%) were not aware that frequent use of enemas can cause CKD, that eating too much red meat is a risk factor for CKD, that chewing khat can cause CKD, and that long-term use of statins are risk factors associated with CKD, respectively, as they responded ‘no/I don’t know’ to these questions and thus demonstrated poor awareness of risk factors associated with CKD.

Appendix A (see Appendix A) shows the participants’ responses to questions assessing their awareness of the risk factors associated with CKD. Just over half of participants (53.2%, and 65%) were aware that uremia, and fluid build-up are complications associated with CKD respectively, as they responded ‘yes to these questions. For the majority of the questions in this section, a very high percentage of participants (81.8%, 80.7%, 79.3%, and 81.8%) were unaware that angina pectoris, sleep disturbances, infertility, and coagulopathy respectively, are some of the complications associated with CKD having responded ‘no/I don’t know’ to the aforementioned questions and consequently, demonstrated poor awareness of complications associated with CKD.

Table 2 depicts the results of cross-tabulations between participants’ socio-demographics and knowledge categories Participants knowledge was significantly associated with being a student or being employed (Government/private employee) (χ^2^ = 29.90; *p* < 0.001), having completed graduate studies (χ^2^ = 63.86; *p* < 0.001), residing in urban areas (χ^2^ = 138.62; *p* < 0.001), belonging to the age group (18–39 years), and having no co-morbidities (χ^2^ = 24.55; *p* < 0.001). Participants’ gender and nationality did not yield any statistically significant results.

Table 3 shows the categorization of the knowledge and awareness scores. By making use of Bloom’s cut-off point criteria [29,30], any score below 60% of the total score was categorized as poor, whereas scores in the range of 60–80% of the total score were categorized as moderate, and scores above 80% of the total score were categorized as good [22]. For questions assessing participants’ knowledge about CKD, it was observed that very few participants (27.3%) had good knowledge, while the remaining either had moderate (36.6%) (or) poor knowledge (36.1%) regarding CKD. With regards to the questions assessing the participants’ awareness of the risk factors and the complications associated with CKD, it was noticed that a very small percentage of participants (7.5% and 9.3%) respectively had good awareness of the risk factors and the complications associated with CKD, respectively. The vast majority (68.4% and 81.8%) had poor awareness of risk factors and complications associated with CKD, respectively. The above results demonstrate that a large percentage of the present sample neither had good knowledge about CKD nor were they aware of the risk factors as well as the complications associated with CKD.

Table 4 elucidates the results of the correlation between the scores of different scales. It was noted that the knowledge score and the awareness of risk factor score (r = 0.42; *p* < 0.01) and the awareness of complications score (r = 0.25; *p* < 0.01), were positively and significantly correlated. A very strong positive and significant correlation (r = 0.62; *p* < 0.01), was also noted between the awareness of risks factors score and the awareness of complications score. This suggests that, as the score on one scale increases, so does the score on the other scales.

Table 5 depicts the results of multivariate binary logistic regression. It was observed in the present study that gender and nationality did not have any statistically significant effect. It was also seen that students were 16.48 times more likely to have appropriate knowledge about CKD in comparison with those who are unemployed (AOR: 16.48 95% CI; 1.30–208.57; *p* < 0.05). Participants who had only completed their primary school were 0.10 times less likely to possess appropriate knowledge about CKD as compared to those who had completed their graduation. (AOR: 0.10 95% CI; 0.03–0.32; *p* < 0.01). We also noticed that the participants living in rural areas were 0.01 times less likely to have appropriate knowledge as compared to those living in urban areas (AOR: 0.01 95% CI; 0.006–0.04; *p* < 0.001). Similarly, those participants without any co-morbidities were 0.39 times less likely to have appropriate knowledge about CKD when compared with those participants with comorbidities (AOR: 0.39 95% CI; 0.19–0.78; *p* < 0.01). 

## 4. Discussion

The present study assessed the knowledge of the general population of Jazan Province about CKD. It also assessed the participant’s awareness of the risk factors associated with CKD as well as the participant’s awareness of the complications associated with CKD.

Overall, we noticed that 27.8% of participants had good knowledge about CKD, but a very high percentage (68.4%) of participants were unaware of the risk factors associated with CKD and 81.8% of participants were unaware of the complications associated with CKD. We noticed that 29.2% of the sample with good knowledge were students. These findings are similar to the results of a cross-sectional study conducted in Rwanda and which found a little more than one half of university students surveyed had a strong understanding of kidney illness [31]. A telephonic study conducted in Hong Kong revealed respondents had a high degree of knowledge, with the majority aware of renal function and dietary sodium as risk factors. These findings are in contrast with our study wherein we only found 27.8% of participants to have good knowledge about CKD. Nearly one half also understood that hypertension and diabetes were significant risk factors for CKD [12]. The findings concur with the findings of our study wherein we also noticed that more than half were aware that diabetes (66.6%) and hypertension (69.8%) were risk factors associated with CKD. Caregivers in Ethiopia were reported to have adequate knowledge about kidney function and the risk factors [32]. In contrast, studies showed the general population has sparse knowledge about CKD [13,33,34,35,36]. Our study population included the general population and our findings are similar to the above findings wherein the general population showed a very low level of knowledge (27.8%) of CKD. A study conducted in Nigeria on non-medical third and fourth-year students found that over one third had little or no understanding of renal disease [34]. The findings are in stark contrast to our findings wherein we noticed that 12.6% of the students had poor knowledge about CKD. In contrast, an equal proportion had some knowledge, and a quarter had excellent knowledge [12]. Our study also noticed comparable findings wherein 29.9% of students had moderate knowledge and 29.2% of students had good knowledge about CKD.

Our study also noticed a positive and significant correlation between the knowledge score and the awareness of risk factor score (r = 0.42; *p* < 0.01). A significant positive correlation was also seen between the knowledge score and the awareness of complications score (r = 0.25; *p* < 0.01). A very strong positive and significant correlation (r = 0.62; *p* < 0.01), was also noted between the awareness of risks factors score and the awareness of complications score. These findings suggest that an increase in one score leads to an increase in the other scores. We can also infer that participants with good knowledge about CKD would have a better awareness of the risk factors as well as the complications associated with CKD. It is also noteworthy that a lack of knowledge can result in the neglect of early warning signs associated with CKD [36]. However, our results are in contrast with the findings of a Malaysian study which found that most respondents with poor knowledge (69.9%) had a positive attitude (68.9%) and also had good practice (88.3%) to reduce the risk of CKD [36]. A survey of 740 outpatients in Jordan found half of the participants had incorrect information on signs and symptoms of CKD, despite exhibiting appropriate knowledge about the disease [33].

Over two-thirds of our participants knew that untreated diabetes and hypertension can be risk factors for CKD. The awareness was higher than another reported study carried out in 2014, which reported that only 12.7% of Iranians were aware that untreated diabetes was a risk factor for CKD, while only 14.4% were aware that untreated hypertension was also a risk factor for CKD [32]. Nearly three-quarters (75.7%) of the respondents were aware that over-use of non-steroidal anti-inflammatories should be avoided.

However, nearly half (43.4%) of our participants were unaware that smoking could be a risk factor for CKD, and more than three-quarters (79.1%) of our respondents were unaware that chewing khat could cause damage to the kidneys. It was earlier reported that chewing khat every day has a detrimental impact on the kidneys [37], and cigarette smoking is an independent risk factor for CKD [38]. As the participants lacked awareness of some vital facets, it is essential to devise and impart health awareness programs on critical issues. For example, the Center for Disease Control recommends regular checkups for individuals with high blood pressure and high glucose levels to help prevent or manage CKD [39]. These variables can be controlled, hence, regular testing among the high-risk group can be a cost-effective intervention. Furthermore, to reduce the course of CKD, counseling on the regular use of antihypertensive medications and avoiding kidney injury can be helpful. Moreover, patients may postpone seeking care and therapy because they lack the requisite CKD knowledge to identify risk factors and signs and symptoms [40].

Adopting ineffective preventive strategies can be threatening to health. For example, more than three-quarters of our participants felt that the use of herbal medicines could cause kidney failure. While this might not be entirely true, more public education material on natural remedies can help in improving their understanding. A study conducted in Tanzania has reported the use of traditional healers and traditional remedies to treat CKD [41]. Other studies also reflected the use of spiritual means or urine therapy for healing CKD [11,39]. Krishnamurthy et al. (2021) also proposed a machine learning model for CKD, which could be a very useful tool for predicting the trends of CKD in the general population. The proposed model can also allow the close monitoring of individuals at risk of CKD, along with its early detection. This can then be used by policymakers for the appropriate allocation of resources, patient-centered management, and efficient educational initiatives to improve the knowledge about CKD, along with creating awareness of the risks and complications associated with CKD [42]. The present study found people staying in rural areas to potentially have less knowledge than those participants staying in urban areas. These findings are similar to Jafar et al. (2020), who also identified living in rural areas as a significant barrier to accessing proper care for CKD and hence could also be a barrier to attaining CKD-related knowledge [43]. In the present study, we did not include the source of participants’ knowledge, but a very similar study conducted by Alshahrani et al., while assessing the public awareness and perception of chronic kidney disease and its risk factors in the southern region of Saudi Arabia, reported that the most common sources of knowledge were social media, mass media, work, books, health education campaigns, and others [44].

To the best of the authors’ knowledge, the present study is the first in Jazan Province to assess not only the knowledge about CKD in the general population but also to assess their awareness of the potential risk factors and the complications associated with CKD.

The potential limitations of the present study include its cross-sectional nature, which does not make it possible to study the causal relationship. There is a substantial risk of selection bias in a volunteer online survey. Taking into consideration the strict COVID-19 restrictions in place, a web-based survey seemed the only plausible and safe method of data collection. Test–retest reliability of the questionnaire couldn’t be confirmed due to the fact that data was collected online and tracing the same individuals who have initially responded to the questionnaire and getting them to provide their responses again was a bit cumbersome. Moreover, the invitation link to participate in the survey was distributed on social media and elderly individuals who do not have internet access, are not active on social media, and are without a proper understanding of Google Forms, might have been at a disadvantage to participate in the study. Furthermore, with convenience sampling, the present study cannot claim to be representative of the larger population of Jazan province. Additionally, the snowball convenience sampling technique might have also led to probable selection bias and self-report surveys are always prone to recall bias. The present study also did not assess the formal/informal sources of information used to attain knowledge about CKD in the study population.

## 5. Conclusions

The results of the present study demonstrate that a large percentage of the present sample neither had good knowledge about CKD nor were they aware of the risk factors and complications associated with CKD. This warrants an urgent need for the meticulous planning, organizing, and conducting of educational initiatives and programs for the general population with a special emphasis on CKD to substantially improve their knowledge and to create general awareness amongst them of the risk factors and complications associated with CKD. Moreover, community pharmacists should also encourage their customers to undergo periodic screening to minimize the risk of undiagnosed CKD.

## Figures and Tables

**Table 1 healthcare-10-01377-t001:** Sociodemographic characteristics (n = 440).

Characteristics	N (%)
**Age (years)**	
18–39	316 (71.8)
40–59	124 (28.2)
**Gender**	
Male	286 (65.0)
Female	154 (35.0)
**Nationality**	
Saudi	404 (91.8)
Non-Saudi	36 (8.2)
**Education**	
Completed primary school	55 (12.5)
Completed secondary school	133 (30.2)
Completed Graduate studies	252 (57.3)
**Occupation**	
Student	103 (23.4)
Employed (private or government)	220 (50.0)
Unemployed	95 (21.6)
Self-employed	22 (5.0)
**Residency**	
Rural	140 (31.8)
Urban	300 (68.2)
**Co-morbidities**	
No	312 (70.9)
Yes	128 (29.1)
**Medical History** **Non-communicable diseases**	
Diabetes	11(2.5)
Hypertension	9 (2.0)
Heart Diseases	14 (3.2)
COPD	3 (0.7)
Kidney diseases	3 (0.7)
Obesity	60 (13.6)
None	312 (70.9)
Diabetes and hypertension	11(2.5)
Diabetes, hypertension and obesity	2 (0.5)
Diabetes and obesity	3 (0.7)
Heart diseases and COPD	4 (0.9)
Kidney diseases and obesity	1 (0.2)
Heart diseases and kidney diseases	1 (0.2)
COPD and obesity	1 (0.2)
Hypertension and heart diseases	2 (0.5)
COPD and kidney diseases	1 (0.2)
Heart diseases and obesity	2 (0.5)
**Characteristics**	**N (%)**
None	411 (93.4)
Other viral infections	10 (2.3)
Covid-19	12 (2.7)
Dengue and other viral infections	1 (0.2)
Tuberculosis and Covid-19	1 (0.2)
Dengue	1 (0.2)
Dengue and Covid-19	4 (0.9)

**Table 2 healthcare-10-01377-t002:** Cross-tabulations—Participants’ socio-demographics and knowledge categories.

Variable	Poor Knowledge	Moderate Knowledge	Good Knowledge	Frequency (Percentage)	χ^2^	*p*-Value
**Gender**						
Male	96 (60.4%)	107 (66.5%)	83 (69.2%)	286 (65.0%)	2.56	0.28
Female	63 (39.6%)	54 (33.5%)	37 (30.8%)	154 (35.0%)		
**Nationality**						
Saudi	148 (93.1%)	147 (91.3%)	109 (90.8%)	404 (91.8%)	0.55	0.76
Non-Saudi	11 (6.9%)	14 (8.7%)	11 (9.2%)	36 (8.2%)		
**Employment Status**						
Student	20 (12.6%)	48 (29.8%)	35 (29.2%)	103 (23.4%)	29.90	*p < 0.001*
Employed (Government/Private)	82 (51.6%)	79 (49.1%)	59 (49.2%)	220 (50.0%)		
Unemployed	41 (25.8%)	28 (17.4%)	26 (21.7%)	95 (21.6%)		
Self-employed	16 (10.1%)	6 (3.7%)	0 (0.0%)	22 (5.0%)		
**Educational Qualification**						
Completed Primary School	38 (23.9%)	14 (8.7%)	3 (2.5%)	55 (12.5%)	63.86	*p < 0.001*
Completed Secondary School	66 (41.5%)	43 (26.7%)	24 (20.0%)	133 (30.2%)		
Completed Graduate Studies	55 (34.6%)	104 (64.6%)	93 (77.5%)	252 (57.3%)		
**Residence of current location**						
Rural	105 (66.0%)	28 (17.4%)	7 (5.8%)	140 (31.8%)	138.62	*p < 0.001*
Urban	54 (34.0%)	133 (82.6%)	113 (94.2%)	300 (68.2%)		
**Age Group**						
18–39 years	68 (42.8%)	148 (91.9%)	100 (83.3%)	316 (71.8%)	106.32	*p < 0.001*
40–59 years	91 (57.2%)	13 (8.1%)	20 (16.7%)	124 (28.2%)		
**Comorbidities**						
No	135 (84.9%)	105 (65.2%)	72 (60.0%)	312 (70.9%)	24.55	*p < 0.001*
Yes	24 (15.1%)	56 (34.8%)	48 (40.0%)	128 (29.1%)		

**Table 3 healthcare-10-01377-t003:** Level of knowledge about chronic kidney disease, awareness of the risk factors, and awareness of the complications associated with CKD.

Knowledge Score out of 20	Knowledge Score in Percentage	Level of Knowledge
<12	<60%	Poor—159 (36.1%)
12–16	60–80%	Moderate—161 (36.6%)
>16	>80%	Good—120 (27.3%)
**Score of** **Level of awareness of risk factors out of 25**	**Awareness score in percentage**	**Level of awareness**
<15	<60%	Poor—301 (68.4%)
15–20	60–80%	Moderate—106 (24.1%)
>20	>80%	High—33 (7.5%)
**Assessment of complications score**	**Assessment score in percentage**	**Level of assessment**
<12	<60%	Poor—360 (81.8%)
12–16	60–80%	Moderate—39 (8.9%)
>16	>80%	High—41 (9.3%)

**Table 4 healthcare-10-01377-t004:** Correlation between different scores.

	Pearson’s Correlation Coefficient ‘r’	*p*-Value
Knowledge Score vs. Awareness of risks factors Score	0.42	*p < 0.01*
Knowledge Score vs. Awareness of complications Score	0.25	*p < 0.01*
Awareness of risks factors Score vs. Awareness of complications Score	0.62	*p < 0.01*

**Table 5 healthcare-10-01377-t005:** Odds of having knowledge related to CKD with selected sample characteristics.

Determinant	Adjusted Odds Ratio (AOR)	95% CI		*p*-Value
		Lower	Upper	
**Gender**				
Male	1.17	0.64	2.12	0.62
Female				*Ref*
**Nationality**				
Saudi	0.81	0.29	2.22	0.68
Non-Saudi				*Ref*
Education				
Education				*p* < 0.001
Completed Primary School	0.10	0.03	0.32	*p* < 0.001
Completed Secondary School	0.43	0.23	0.78	0.005 *p* < 0.01
Completed Graduate studies				*Ref*
**Employment Status**				
Employment Status				0.001 *p* < 0.01
Student	16.48	1.30	208.57	0.03 *p* < 0.05
Employed	2.92	0.25	33.79	0.39
Self-employed	3.59	0.30	43.13	0.31
Unemployed				*Ref*
**Location of current residence**				
Rural	0.01	0.006	0.04	*p* < 0.001
Urban				*Ref*
**Presence of co-morbidities**				
Co-morbidities Absent	0.39	0.19	0.78	0.007 *p* < 0.01
Co-morbidities present				*Ref*

## Data Availability

Data of the present study can be made applicable upon reasonable request to the corresponding author (S.A.).

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
