# Peer review of "Public Awareness of Chronic Kidney Disease in Jazan Province, Saudi Arabia—A Cross-Sectional Survey"

_healthcare, 2022, doi:10.3390/healthcare10081377_

Round 1
Reviewer 1 Report
The authors have subsantially improved the manuscript according to the comments by the rewievers. I have only a few minor comments:
- results are very long, with many large tables that are not so reader-friendly; I suggest putting some of them into supplementay material
- Table 2 can be removed and incorporated in text in Methods section
- study limitations don't need individual caption (5.) . they can be incorporated into Discussion
Author Response
Study Title: Public awareness of chronic kidney disease in Jazan province, Saudi Arabia - A cross sectional survey
Study I.D: healthcare-1758845.
07-07-22
Dear Editor,
I, the corresponding author, on behalf of all the co-authors would like to express our utmost thankfulness and most sincere gratitude for providing us with another opportunity of submitting a revised version of our manuscript.
We also thank the reviewers very much for their valuable comments/suggestions which have hugely helped us improving our manuscript. All their kind comments/suggestions are much appreciated. All their kind comments/queries have now been addressed systematically in the response letter and their kind suggestions have now been duly incorporated into the manuscript as well. The revised text has duly been highlighted in yellow.
We hope that we have done enough for the paper to be acceptable for publication in Healthcare.
Thanking you very much indeed once again.
Yours Sincerely,
Dr. Saeed Alshahrani
(Corresponding Author)
RESPONSES TO REVIEWER 1
Dear Reviewer 1,
We are extremely grateful for the invaluable input provided. This has immeasurably improved the quality of our manuscript. All the comments have now been responded to and the manuscript modified accordingly.
The revised text in the manuscript has been highlighted in yellow.
|
We are extremely appreciative of the kind Reviewer 1 for the most invaluable comments/suggestions provided. |
|
|
Comment |
Response |
|
The authors have subsantially improved the manuscript according to the comments by the rewievers. I have only a few minor comments:
|
We are exceedingly grateful for the kind comments. |
|
- results are very long, with many large tables that are not so reader-friendly; I suggest putting some of them into supplementay material |
As suggested by the kind Reviewer table 2-4 have been added as supplementary materials. |
|
- Table 2 can be removed and incorporated in text in Methods section |
In line with the suggestions of the Kind Reviewer the results of Table 2 have now been incorporated into the methods section. |
|
- study limitations don't need individual caption (5.) . they can be incorporated into Discussion |
Caption 5 from the limitations section has now been duly removed. |
Thank you very much indeed for your invaluable comments/suggestions and your time.
Yours Sincerely,
Dr. Saeed Alshahrani
(Corresponding Author)

Reviewer 2 Report
1. This study data was collected by Google Forms, making use of a validated 73-item self-report survey on diverse platforms of social media. The participant’s ages range from 18 to 59 years, and a mean age of 32.66 years (SD ± 10.83), which may be most in the young age students groups. “18-39” n=316 (71.8%)
2. The manuscript results of the research are understandable and predictable, and there are no very special findings. I don't think it provides a new experimental design or new creative finding.
3. This research does not provide evidence of ethics review, the “Ethics approval and consent to participate” should be described in the manuscript, about IRB (Institutional Review Board) / IEC (Independent Ethics Committee) approval, or the data is applied for or authorized to use.
4. Lack of more innovative research design and research method application in this study. It is a cross-sectional study; the publication risk of this article has been described in the limitation part of this manuscript.
5. This article does not provide enough value as a published commentary to help readers understand its significance and place in the literature or contribute beyond what is already known in other references.
Author Response
Study Title: Public awareness of chronic kidney disease in Jazan province, Saudi Arabia - A cross-sectional survey
Study I.D: healthcare-1758845.
07-07-22
Dear Editor,
I, the corresponding author, on behalf of all the co-authors would like to express our utmost thankfulness and most sincere gratitude for providing us with another opportunity of submitting a revised version of our manuscript.
We also thank the reviewers very much for their valuable comments/suggestions which have hugely helped us improving our manuscript. All their kind comments/suggestions are much appreciated. All their kind comments/queries have now been addressed systematically in the response letter and their kind suggestions have now been duly incorporated into the manuscript as well. The revised text has duly been highlighted in yellow.
We hope that we have done enough for the paper to be acceptable for publication in Healthcare.
Thanking you very much indeed once again.
Yours Sincerely,
Dr. Saeed Alshahrani
(Corresponding Author)
RESPONSES TO REVIEWER 2
Dear Reviewer 2,
Thank you very much indeed for the invaluable input and your kind comments. These have helped us enormously in proving the overall quality of our manuscript. All your comments/suggestions are most welcome and have now been systematically addressed and the manuscript modified accordingly.
The revised text in the manuscript has been highlighted in yellow.
|
We thank the kind Reviewer for the invaluable comments. |
|
|
Comment |
Response |
|
1. This study data was collected by Google Forms, making use of a validated 73-item self-report survey on diverse platforms of social media. The participant’s ages range from 18 to 59 years, and a mean age of 32.66 years (SD ± 10.83), which may be most in the young age students groups. “18-39” n=316 (71.8%) |
As we are an academic institution most of the independent data collectors tend to first collect the data from the university students and then proceed to the general population. This could be a probable reason for a very high proportion of the study sample to be young. |
|
2. The manuscript results of the research are understandable and predictable, and there are no very special findings. I don't think it provides a new experimental design or new creative finding. |
All the authors and the independent data collectors have diligently and earnestly conducted this study with a lot of devotion and dedication. We honestly and sincerely believe that our research findings have scientific validity and social value wherein by making use of our research findings (which clearly shows that very low percentage of sample (27.3%., 7.5%, 9.3%) had good knowledge, high level of awareness of the risk factors and about the complications associated with CKD respectively. We assiduously feel that the data generated from the present study can be used to plan, design and implement thorough, meticulous and extensive CKD educational initiatives concentrating on improving the general knowledge of CKD along with improving the overall awareness of the public towards the risk factors as well as the complications associated CKD. |
|
3. 3. This research does not provide evidence of ethics review, the “Ethics approval and consent to participate” should be described in the manuscript, about IRB (Institutional Review Board) / IEC (Independent Ethics Committee) approval, or the data is applied for or authorized to use. |
The present study was conducted only after it was thoroughly reviewed and subsequently approved by the Institutional Research Review and Ethics Committee of Jazan University (Letter number REC 42/1/127 dated 5th May 2021). Informed consent was taken by requesting the study participants to select 'Yes' to a mandatory question seeking their consent. Providing informed consent was imperative for the study participants to proceed to other sections of the questionnaire. A 'No' answer automatically ended the survey, with the corresponding response being considered a dropout. |
|
4. Lack of more innovative research design and research method application in this study. It is a cross-sectional study; the publication risk of this article has been described in the limitation part of this manuscript. |
Even though the present study may not have deployed a more innovative research design and research methodology but we genuinely and solemnly feel that the findings of the present study have scientific validity and social value and the findings can potentially be used to used to plan, design and implement thorough, meticulous and extensive CKD educational initiatives that which have a massive impact on improving the overall knowledge about CKD and its associated risk factors and complications. |
|
5. 5. This article does not provide enough value as a published commentary to help readers understand its significance and place in the literature or contribute beyond what is already known in other references. |
We would kind-heartedly like to bring to the attention of the kind reviewer to the information already provided above which kindly shows the scientific validity and social value of the present study. |
Thank you very much indeed for your time.
Yours Sincerely,
Dr. Saeed Alshahrani
(Corresponding Author)

Reviewer 3 Report
1. Line 59 should be CKD
2. Line 60 awareness among whom? Also, ref. 8 has been split into 8 and 9. This changes all the references after 8 in the manuscript.
3. Sample size calculation should be brief. The exact formulas and assumptions are not necessary.
4. Spell check is needed. For e.g., Heart is mispelled as Herat in Table 1.
5. The study has interesting findings and implications for rural and lesser educated populations that should be discussed.
6. Some parts of the discussion could be moved to the background section to focus on the implications of the study.
7. One area that was not well covered was how people in Saudi Arabia gain knowledge about CKD. What are the formal/informal sources of information?
Author Response
Study Title: Public awareness of chronic kidney disease in Jazan province, Saudi Arabia - A cross sectional survey
Study I.D: healthcare-1758845.
07-07-22
Dear Editor,
I, the corresponding author, on behalf of all the co-authors would like to express our utmost thankfulness and most sincere gratitude for providing us with another opportunity of submitting a revised version of our manuscript.
We also thank the reviewers very much for their valuable comments/suggestions which have hugely helped us improving our manuscript. All their kind comments/suggestions are much appreciated. All their kind comments/queries have now been addressed systematically in the response letter and their kind suggestions have now been duly incorporated into the manuscript as well. The revised text has duly been highlighted in yellow.
We hope that we have done enough for the paper to be acceptable for publication in Healthcare.
Thanking you very much indeed once again.
Yours Sincerely,
Dr. Saeed Alshahrani
(Corresponding Author)
RESPONSES TO REVIEWER 3
Dear Reviewer 3,
Thank you very much indeed for the invaluable input and your kind comments. These have helped us enormously in proving the overall quality of our manuscript. All your comments/suggestions are most welcome and have now been systematically addressed and the manuscript modified accordingly.
The revised text in the manuscript has been highlighted in yellow.
|
We are exceedingly grateful to the kind Reviewer 3 for the most valuable comments/suggestions provided. |
|
|
Comment |
Response |
|
1. Line 59 should be CKD |
In accordance with the suggestion made by the kind Reviewer this has now been updated. |
|
2. Line 60 awareness among whom? Also, ref. 8 has been split into 8 and 9. This changes all the references after 8 in the manuscript. |
This sentence have now been updated to include “Hence, the present study was designed and conducted to assess the knowledge of CKD, and also to assess the awareness of the risk factors as well as the complications associated with CKD amongst the general population of Jazan Province.” References 8 and 9 have now been merged and all the subsequent references have now been duly updated. |
|
3. Sample size calculation should be brief. The exact formulas and assumptions are not necessary. |
Sample size calculation section has now been duly concised. |
|
4. Spell check is needed. For e.g., Heart is mispelled as Herat in Table 1. |
A thorough and an extensive review of the manuscript was carried out to identify and remove typographical errors, mis-spellings and incorrectly spelled words. |
|
5. The study has interesting findings and implications for rural and lesser educated populations that should be discussed. |
The findings of the study have now been enriched with discussion on how living in rural areas might have a potential impact on the participants’ knowledge of CKD. |
|
6. Some parts of the discussion could be moved to the background section to focus on the implications of the study. |
In accordance with the suggestions of the kind Reviewer the initial part of the discussion has now been moved to the background. |
|
7. One area that was not well covered was how people in Saudi Arabia gain knowledge about CKD. What are the formal/informal sources of information? |
Now that the study has already been concluded, we would most definitely include the formal/informal sources of participants’ knowledge in our upcoming KAP studies. Inability to do so has duly been highlighted in the limitations section. |
Thank you very much indeed, we are much obliged for your valuable comments/suggestions and your time.
Yours Sincerely,
Dr. Saeed Alshahrani
(Corresponding Author)

Round 2
Reviewer 2 Report
The authors revised the manuscript and provided the question's explanations.